# ON THE EFFECTIVENESS OF TASK GRANULARITY FOR TRANSFER LEARNING

## ABSTRACT

We describe a DNN for video classification and captioning, trained end-to-end, with shared features, to solve tasks at different levels of granularity, exploring the link between granularity in a source task and the quality of learned features for transfer learning. For solving the new task domain in transfer learning, we freeze the trained encoder and fine-tune an MLP on the target domain. We train on the Something-Something dataset with over $220,000$ videos, and multiple levels of target granularity, including $50$ action groups, $174$ fine-grained action categories and captions. Classification and captioning with Something-Something are challenging because of the subtle differences between actions, applied to thousands of different object classes, and the diversity of captions penned by crowd actors. Our model performs better than existing classification baselines for Something-Something, with impressive fine-grained results. And it yields a strong baseline on the new Something-Something captioning task. Experiments reveal that training with more fine-grained tasks tends to produce better features for transfer learning.

## 1 INTRODUCTION

Common-sense video understanding entails fine-grained recognition of actions, objects, spatial temporal relations, and physical interactions, arguably well beyond the capabilities of current techniques. A general framework will need to discriminate myriad variations of actions and interactions. For example, we need to be able to discriminate similar actions that differ in subtle ways, e.g., *'putting a pen beside the cup'*, *'putting the pen in the cup'*, or *'pretending to put the pen on the table'*. Similarly, one must be able to cope with diverse actors and object classes. Such generality and fine-grained discrimination are key challenges to video understanding. In contrast to current approaches to action recognition and captioning on relatively small corpora, with coarse-grained actions, this paper considers fine-grained classification and captioning tasks on large-scale video corpora. Training is performed on the Something-Something dataset (Goyal et al. (2017)), with $174$ fine-grained action categories and thousands of different objects, all under significant variations in lighting, viewpoint, background clutter, and occlusion.

We describe a DNN architecture comprising a 2-channel convolutional net and an LSTM recurrent network for video encoding. A common encoding is shared for classification and caption generation. The resulting network performs better than baseline action classification results in Goyal et al. (2017), and it gives impressive results on an extremely challenging fine-grained captioning task. We also demonstrate the quality of learned features through transfer learning from Something-Something features to a kitchen action dataset. The main contributions in the paper include:

1. **Explore the link between label granularity and feature quality**: We exploit 3 levels of granularity in sth-sth-v2, namely, action groups, action categories, and captions. Experiments show that more fine-grained labels yield richer features (see Table 3 and Fig. 4).

2. **Baselines for captioning on sth-sth-v2 data**: We note that the captioning task is new for this dataset; the original dataset did not provide captions.

3. **Captioning as a source task for transfer learning**: We show that models trained to jointly perform classification and captioning learn features that transfer better to other tasks (e.g., see Fig. 4). To the best of our knowledge, captioning has, to date, been used as a target task. Our results suggest that captioning is a powerful source task.

4. **20bn-kitchenware**: We introduce a new dataset, ostensibly for video transfer learning.

## 2   VIDEO DATA

Video action classification and captioning have received significant attention for several years, but progress has been somewhat slow in part because of a lack of large-scale corpora. Using web sources (e.g., YouTube, Vimeo and Flickr) and human annotators, larger datasets have been collected (e.g., Kay et al. (2017); Monfort et al. (2018)), but they lack control over pose variations, motion and other scene properties that might be important for learning fine-grained models. More recently, crowd-sourced/crowd-acted data has emerged. This allows targeted video domains, action classes, and control over subtle differences between fine-grained action classes. The first version of Goyal et al. (2017) has $100,000$ videos of human-object interactions, comprising $50$ coarse-grained action groups, decomposed further into $174$ related action categories. The videos exhibit significant diversity in viewing and lighting, objects and backgrounds, and the ways in which actors perform actions. Baseline performance in Goyal et al. (2017) was a correct action classification rate of $11.5\%$, and $36.2\%$ on action groups. Zhou et al. (2017) report $42.01\%$ classification accuracy on Something-Something-V1 action categories. With our architecture we obtain validation accuracy of $38.8\%$.

Something-Something-V2 is larger, with $220,847$ videos of the same $174$ action categories. In addition, each video includes a caption that was authored and uploaded by the crowd actor. These captions incorporate the action class as well as descriptions of the objects involved. That is, the captions mirror the action template, but with the generic placeholder *Something* replaced by the object(s) chosen by the actor. As an example, a video with template action *'Putting [something] on [something]'*, might have a caption *'Putting a red cup on a blue plastic box'*. In a nutshell, this dataset provides different levels of granularity: $50$ coarse-grained action groups, $174$ fine-grained action categories, and even more fine-grained labeling via video captions.

## 3   VIDEO CLASSIFICATION AND CAPTIONING TASKS

Due to the prevalence of datasets like UCF-101 (Soomro et al. (2012)), sports1M (Karpathy et al. (2014)) and more recently Kinetics (Kay et al. (2017)), most research on action classification has focused on models for coarse-grained action classification. In extreme cases, action classes can be can be discriminated from a glimpse of the scene, often encoded in isolated frames; e.g., inferring "soccer play" from the presence of a green field. Even when motion is essential to the action, many existing approaches do well by encoding rough statistics over velocities, directions, and motion positions. Little work has been devoted to the task of representing details of object interactions or how configurations change over time.

Image and video captioning have received increasing attention since the release of captioning data, notably, Microsoft COCO (Chen et al. (2015)) and MSR-VTT (Xu et al. (2016)). One problem with existing captioning approaches is that many datasets implicitly allow models to "cheat", e.g., by generating phrases that are grammatically and semantically coherent, but only loosely related to the fine-grained visual structure. It has been shown, for example, that a language model trained on unordered, individual words (e.g., object-nouns) predicted by a separate NN can compete with captioning model trained on the actual task end-to-end (e.g., Yao et al. (2015); Heuer et al. (2016)). Similarly, nearest neighbor methods have been surprisingly effective (Devlin et al. (2015)).

Captioning tasks, if designed appropriately, should capture detailed scene properties. Labels with more subtle and fine-grained distinctions would directly expose the ability (or inability) of a network to correctly infer the scene properties encoded in the captions. The captions provided with the Something-Something dataset are designed to be sufficiently fine-grained that attention to details is needed to yield high prediction accuracy. For example, to generate correct captions, networks need not only infer the correct actions, but must also learn to recognize different objects and their properties, as well as object interactions.

## 4   APPROACH

We use a modified encoder-decoder architecture with an action classifier applied to the video encoding (see Fig. 1). The decoder or classifier can be switched off, leaving pure classification or captioning models. One can also jointly train classification and captioning models. We train our two-channel architecture to solve four different tasks with different granularity levels:
   - Coarse-grained classification on action groups (CG actions)
   - Fine-grained classification on 174 action categories (FG actions)

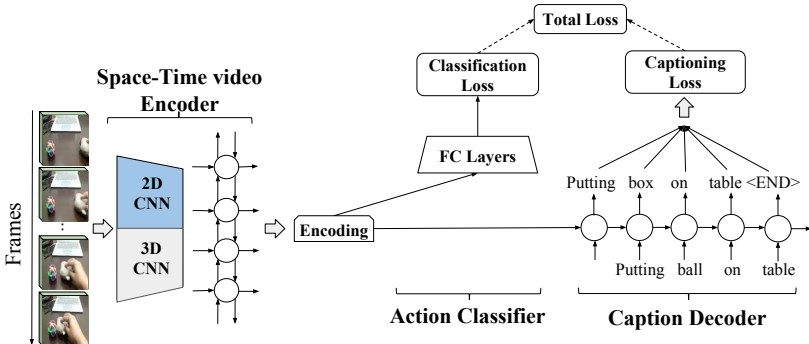

Figure 1: Our model architecture includes a two-channel CNN followed by an LSTM video encoder, an action classifier, and an LSTM decoder for caption generation.

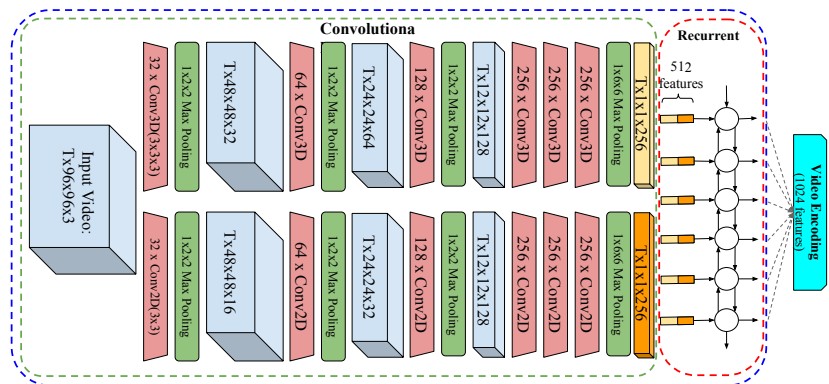

Figure 2: Our encoder includes a two-channel CNN followed by an LSTM for aggregating features.

  - Captioning on simplified objects (SO captions)
  - Fine-grained Captioning with full object placeholders (Captions)
Finally, we investigate the effectiveness of the learned features for transfer learning.

## 4.1 MODIFIED VIDEO ENCODER-DECODER

The *video encoder* receives the input video $v$, and maps it to an embedded representation $h$. Conditioned on $h$, a caption decoder generates the caption $c$, and a classifier predicts the action category. The encoder processes the video with a two-channel convolutional architecture (Fig. 2). A spatial 2D-CNN and a spatiotemporal 3D-CNN are applied in parallel. The 3D-CNN features are used in lieu of a separate module to compute motion (e.g., optical flow) features. The basic building block of each channel is a $3 \times 3 \times 3$ ($3 \times 3$ in 2D-CNN channel) convolution filter with batchnorm and ReLU activation. To aggregate features across time, feature vectors from each channel are combined and fed to a 2-layer bidirectional LSTM. We average these features to get the video encoding, $h$.

The *action classifier* applies a fully-connected layer to the encoding $h$, followed by a softmax layer. Training uses a cross-entropy loss over action categories. The *caption decoder* is a two-layer LSTM. Much like conventional encoder-decoder methods for video captioning (Venugopalan et al. (2014); Donahue et al. (2015); Kaufman et al. (2016) and Sutskever et al. (2014)), our decoder generates captions using a softmax over vocabulary words, conditioned on previously generated words. The loss is the usual negative log-probability of the word sequence:

$$\text{loss}_{\text{captioning}} = - \sum_{i=0}^{N-1} \log p(w^{i+1}|w^{\leq i}, h; \theta). \tag{1}$$

where $w^i$ denotes the $i^{\text{th}}$ word of the caption, $h$ is the video encoding, and $\theta$ denotes model parameters. To optimize speed and memory usage during training, the length of captions generated by the decoder is fixed at 14 words[1]. As is common for encoder-decoder networks, we train with teacher-

---

[1]Less than 1% of Something-Something captions have more than 14 words.

| Models | 3D-CNN Channels | 2D-CNN Channels | Number of Parameters | Validation Accuracy | Test Accuracy |
|---|---|---|---|---|---|
| M(256 − 0) | 256 | 0 | 8.9M | 50.06 | 48.84 |
| M(512 − 0) | 512 | 0 | 24.1M | 51.96 | **51.15** |
| M(384 − 128) | 384 | 128 | 16.2M | 51.11 | 49.96 |
| M(256 − 256) | 256 | 256 | 11.5M | 51.62 | **50.44** |
| M(128 − 384) | 128 | 384 | 10.0M | 50.82 | 49.57 |
| M(0 − 512) | 0 | 512 | 5.8M | 39.78 | 37.80 |
| M(0 − 256) | 0 | 256 | 11.5M | 40.2 | 38.83 |

Table 1: Validation and test accuracy for the pure classification task ($\lambda = 1$), with different numbers of 2D-CNN and 3D-CNN features used for video encoding.

| | Coarse-grained Testing | Fine-grained Testing |
|---|---|---|
| Coarse-grained Training | 57.6 | 41.7 |
| Fine-grained Training | **62.5** | **50.44** |

Table 2: Comparison of classification accuracy of fine-grained and coarse-grained models, tested on fine-grained actions (using action categories) versus coarse-grained actions (using action groups).

forcing (Williams & Zipser (1989)). At test time, the input to the decoder at each time-step is the token generated at the previous time-step (i.e., no teacher forcing). The model is trained end-to-end for classification and captioning with a weighted sum of the classification and captioning losses, i.e.,

$$\text{loss} = \lambda \cdot \text{loss}_{\text{classification}} + (1 - \lambda) \cdot \text{loss}_{\text{captioning}} \qquad (2)$$

With $\lambda = 1$ or $\lambda = 0$, we end up with pure classification and captioning tasks respectively. For other values of $\lambda$, they are trained jointly. The encoder is shared by the action classifier and the caption decoder. The experiments below also compare this joint training regime with models for which the encoder is trained on the classification loss or the captioning loss alone.

## 5 EXPERIMENTS

We train four two-channel models that are trained to solve coarse-grained and fine-grained classification and captioning tasks. In what follows we discuss different tasks in more details.

### 5.1 COARSE- AND FINE-GRAINED CLASSIFICATION

Something-Something provides coarse-grained categories (action groups), each comprising a set of fine-grained actions. We trained a classification model on coarse-grained action groups, using the M(256-256) architecture, with accuracy of $57.6\%$ (see Table 2 (top-left)). Table 1 reports the performance of our model for fine-grained classification. For the pure classification task (with $\lambda = 1$) we consider different numbers of features produced by the 2D (spatial) CNN and the 3D (spatio-temporal) CNN. The total number of features is $512$ in all cases. The results show that the model benefits from 2D and 3D features. The even split M(256-256) provides a good trade-off between performance and model complexity. We therefore use this model below, unless otherwise specified.

**Transferring between Coarse- and Fine-Grained Classification** We can perform coarse-grained classification by mapping predictions from the fine-grained classifier onto the action groups. To this end we sum the probabilities of fine-grained actions belonging to each action group. Interestingly, the resulting accuracy on coarse-grained action groups increases to $62.5\%$. This improvement suggests that fine-trained training provides higher quality features. We also examine to what extent coarse-grained performance accounts for fine-grained accuracy; i.e, how better fine-grained performs compared to chance when conditioned on coarse-grain predictions. For example, conditioned on a predicted action group, if we select the most frequent action within the action group, fine-grained test accuracy would be $23.8\%$. One can also fix the coarse-grained model and train a linear classifier on top of its penultimate features. This yields test accuracy of $41.7\%$, still $8.7\%$ below test performance for the corresponding architecture trained on the fine-grained task, further supporting our contention that training on fine-grained details yields richer features.

### 5.2 CLASSIFICATION BASELINES.

As a baseline, we use ImageNet-pretrained models (Simonyan & Zisserman (2014); He et al. (2015)) on individual frames, to which we add layers. First, we use just the middle frame of the video, with

| Models | Test Accuracy |
|---|---|
| VGG16 + MLP 1024 (single middle frame) | 13.29 |
| VGG16 + MLP 1024 (averaged over 48 frames) | 17.57 |
| VGG16 + LSTM 1024 | **31.69** |
| ResNet152 + MLP 1024 (single middle frame) | 13.62 |
| ResNet152 + MLP 1024 (averaged over 48 frames ) | 16.79 |
| ResNet152 + LSTM 1024 (48 steps) | **28.82** |

Table 3: Classification results on 174 action categories using VGG16 and ResNet152 as frame encoders, along with different strategies for temporal aggregation.

a 2-layer MLP with 1024 hidden units. We also consider a baseline in which we apply this approach to all 48 frames, and then average frame by frame predictions. We experiment with aggregating information over time using a LSTM layer with 1024 units. We report results in Table 3. There is a marked improvement with the LSTM, confirming that this task improves with temporal analysis. Our best baseline was achieved with a VGG architecture, and the test accuracy is close to the best architecture reported to date on Something-Something (e.g., Zhou et al. (2017)).

### 5.3 Captioning with simplified object placeholders.

The ground truth object placeholders in Something-Something video captions (i.e., the object descriptions provided by crowd actors) are not highly constrained. Crowd actors have the option to type in the objects they used, along with multiple descriptive or quantitative adjectives, elaborating shape, color, material or the number of objects involved. Accordingly, it is not surprising that the distribution over object placeholders is extremely heavy-tailed, with many words occurring rarely. To facilitate training we therefore replaced all words that occurred 5 times or less by [Something]. After removing rare words, we are left with 2880 words comprising around 30,000 distinct object placeholders (i.e., different combination of nouns, adjectives, etc).

We consider a simplified task in which we modify the ground truth captions so they only contain one word per placeholder, by keeping the last noun, removing all other words from the placeholders. By substituting the pre-processed placeholders into the action category, we obtain a simplified caption. Table 5 shows an example of the process. The result is a reduced vocabulary with 2055 words. In the spectrum of granularity, captioning with simplified objects can be considered as a middle ground between fine-grained action classification and captioning with full labels.

We train different variations of our two-channel architecture on captions with simplified objects. Table 6 summarizes our results. We observe that the model with an equal number of 2D- and 3D-channels performs best (albeit by a fairly small margin). Also the best captioning model performs best on the classification task. We also evaluate the models using standard captioning metrics: BLEU (Papineni et al. (2002)), ROUGE-L (Lin (2004)) and METEOR (Denkowski & Lavie (2014)).

### 5.3.1 Fine-grained Captioning with full object placeholders.

We also train networks on the full object placeholders. This constitutes the finest level of action granularity. The experimental results are shown in Table 8. They show that, again, the best captioning model also yields the highest corresponding classification accuracy. The Exact-Match accuracy is significantly lower than for the simplified object placeholders, as it has to account for a much wider variety of phrases. The captioning models produce impressive qualitative results with a high degree of approximate action and object accuracy. Some examples are shown in Figure 9n. More examples can be found in the appendix.

To the best of our knowledge there are no baselines for the Something-Something captioning task. To quantify the performance of our captioning models, we count the percentage of generated captions that match ground truth word by word. We refer to this as "Exact-Match Accuracy". This is a challenging metric as the model is deemed correct only if it generates the entire caption correctly. If we use the action category predicted by model M(256-256), trained for classification, and replace all occurrences of [something] with the most likely object string conditioned on that action class, the Exact-Match accuracy is 3.15%. The same baseline for simplified object placeholders is 5.69%. We also implemented a conventional encoder-decoder model for captioning 4.

| Models | BLEU@4 | ROUGE-L | METEOR | Exact-Match Accuracy | Classification Accuracy |
|---|---|---|---|---|---|
| VGG16+LSTM | 31.83 | 52.22 | 24.79 | 3.13 | 31.69 |
| Resnet152+LSTM | 31.93 | 51.76 | 24.89 | 3.25 | 28.82 |

Table 4: Captioning baselines using a conventional encoder-decoder architecture

| Video ID | 81955 |
|---|---|
| Action Group | Holding [something] |
| Action Category | Holding [something] in front of [something] |
| Somethings | "a blue plastic cap", "a men's short sleeve shirt" |
| Simplified somethings's | "cap", "shirt" |
| Simplified-object Caption | Holding cap in front of shirt |
| Full Caption | Holding a blue plastic cap in front of a men short sleeve shirt |

Table 5: An example of annotation file for a Something-Something video

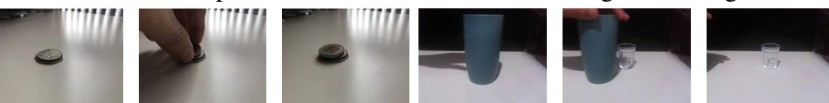

Figure 3: Captioning examples:
**Model Outputs:** [Piling coins up], [Removing mug, revealing cup behind].
**Ground Truths:** [Stacking coins], [Removing cup, revealing little cup behind].

**Training settings**    In all our experiments we use frame rate of $12fps$. During training we randomly pick 48 consecutive frames. For videos with less than 48 frames, we replicate the first and last frames to achieve the intended length. We resize the frames to $128 \times 128$, and then use random cropping of size $96 \times 96$. For validation and testing, we use $96 \times 96$ center cropping. We optimize all models using Adam, with an initial learning rate of 0.001.

While the captioning task theoretically entails action classification, we found that our two-channel networks optimized on the pure captioning task do not perform as well as models trained jointly on classification and captioning (see 7). By coarsely tuning the hyper-parameter $\lambda$ empirically, we found $\lambda = 0.1$ to work well and fix it at this value for the captioning experiments below. More specifically, we first train with a pure classification loss, by setting $\lambda = 1$, and subsequently introduce the captioning loss by gradually decreasing $\lambda$ to 0.1.

# 6    TRANSFER LEARNING

One astonishing property of neural networks is their ability to learn representations transfer well to other tasks (e.g., Donahue et al. (2013); Sharif Razavian et al. (2014)). A distinguishing feature of the ImageNet task, which likely contributes to its potential for transfer learning, is the dataset size and the variety of fine-grained discriminations required. In what follows we explore transfer learning performance as a function of course task granularity.

We introduce *20bn-kitchenware*, a few-shot video classification dataset with 390 videos over 13 action categories (see Fig. 4). It contains video clips of manipulating a kitchen utensil for roughly 4 seconds and was designed to capture fine-grained actions with subtle differences. For each utensil $X \in \{$fork, spoon, knife, tongs$\}$, the target label belongs to one of 3 actions, *"Using X"*, *"Pretending to use X"* or *"Trying but failing to use X"*. In addition to these 12 action categories, we also include a fall-back class labeled *"Doing other things"*.

We further encourage the model to pay attention to visual details by including unused 'negative' objects. The last row of Fig. 4 shows one example; the target label indicates a manipulation of tongs, but the clip also contains a spoon with an egg in it. Given the limited amount of data available for training[2], the action granularity and the presence of negative objects, we hypothesize that only models that have some understanding of physical world properties will perform well. We will release 20bn-kitchenware upon publication of this paper.

---

[2]130 samples – the rest are used for validation and testing.

| Models | BLEU@4 | ROUGE-L | METEOR | Exact-Match Accuracy | Classification Accuracy |
|---|---|---|---|---|---|
| M(256 − 0) | 22.75 | 44.54 | 22.40 | 8.46 | 50.64 |
| M(512 − 0) | 23.28 | 45.29 | 22.75 | 8.47 | 50.96 |
| M(384 − 128) | 23.02 | 44.86 | 22.58 | 8.53 | 50.73 |
| M(256 − 256) | 23.04 | 44.89 | 22.60 | **8.63** | **51.38** |
| M(128 − 384) | 22.76 | 44.40 | 22.39 | 8.33 | 50.04 |

Table 6: Performance of our two-channel models with different sizes of channel features on for **simplified objects**. For this task we use ($\lambda = 0.1$). The maximum sequence length is 14.

| Models | Classification Accuracy | Exact-Match Accuracy |
|---|---|---|
| $\lambda = 0$ | 39.78 | 5.96 |
| $\lambda = 0.1$ | **51.32** | **8.63** |

Table 7: Comparing models trained with pure captioning task vs joint captioning and classification. Results are shown for captioning with simplified object placeholders. The test classification accuracy for the pure captioning model was obtained by freezing the video encoder and training a linear regressor on top of penultimate features.

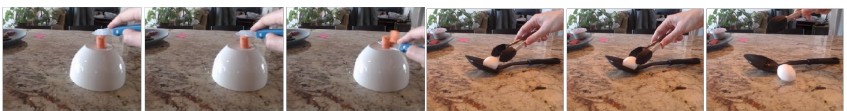

Figure 4: 20bn-kitchenware samples: Using a knife to cut something (left), Trying but failing to pick something up with tongs(right).

### 6.1 PROPOSED BENCHMARK

Our transfer learning experiments considers four two-channel models that have been respectively pre-trained on coarse-grained labels (classification on action groups), on fine-grained labels (classification on 174 action categories), on simplified captions (captioning on fine-grained action categories expanded with single object descriptor) and on template captions (captioning on fine-grained action categories expanded with object descriptors). We also include two neural nets pre-trained on other datasets: a VGG16 network pre-trained on ImageNet, and an Inflated-ResNet34 pre-trained on Kinetics[3].

The overall training procedure remains the same for all models. We freeze each pre-trained model and then train a neural net on top of extracted penultimate features. Independent of the architecture used, we use the pre-trained model to produce 12 feature vectors per second. To achieve this, where necessary we split the input video into smaller clips and apply the pre-trained network on each clip individually[4]. In the simplest case, we pass the obtained features through a logistic regressor and average predictions over time. We also report results for which we classify pre-trained features using an MLP with 512 hidden units as well as a single bidirectional LSTM layer with 128 hidden states. This allows the network to perform some temporal analysis about the target domain.

### 6.1.1 OBSERVATIONS

For each pretrained model and classifier, we evaluate 1-shot, 5-shot and 10-shot performance, averaging scores over 10 runs. Fig. 5 shows the average scores with 95% confidence intervals. The most noticeable findings are:

1. *Logistic Regression vs MLP vs BiLSTM*: Using a recurrent network yields better performance.

2. *Something-Something features vs others*: Our models pre-trained on Something-Something outperform other external models. This is not surprising given the target domain; 20bn-kitchenware samples are, by design, closer to Something-Something samples than ImageNet or Kinetics ones. It is surprising that VGG16 features perform better on 20bn-kitchenware than Kinetics features.

---

[3]https://github.com/kenshohara/3D-ResNets-PyTorch
[4]VGG16 applied to individual frames. For Inflated-ResNet34, video was split into clips of 16 frames.

| Models | BLEU@4 | ROUGE-L | METEOR | Exact Match Accuracy | Classification Accuracy |
|--------|--------|---------|--------|----------------------|-------------------------|
| M(256 − 0) | 16.87 | 40.03 | 19.13 | 3.33 | 50.48 |
| M(512 − 0) | 16.92 | 40.54 | 19.26 | 3.56 | 49.81 |
| M(384 − 128) | 17.99 | 41.82 | 20.03 | **3.80** | **50.92** |
| M(256 − 256) | 17.61 | 41.28 | 19.69 | 3.76 | 50.56 |
| M(128 − 384) | 16.80 | 39.98 | 19.11 | 3.61 | 49.24 |

Table 8: Performance of captioning models with different sizes of channel features on full object placeholders. For this task we use $(\lambda = 0.1)$. The maximum sequence length is 14.

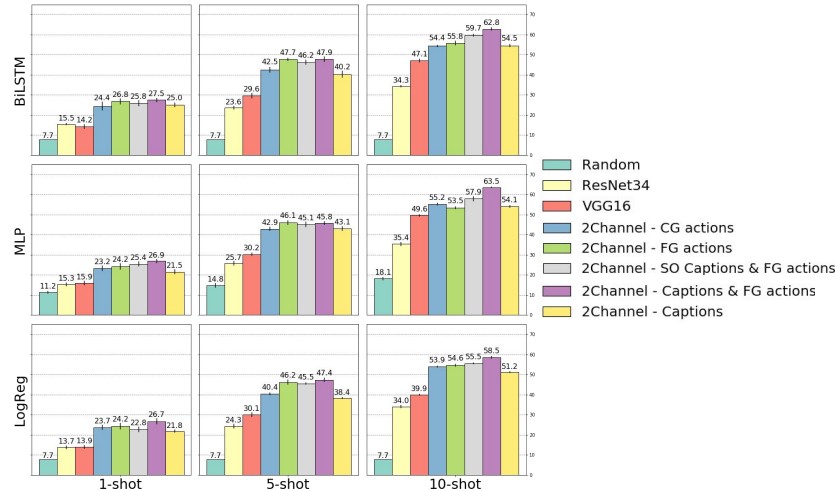

Figure 5: 20bn-kitchenware transfer learning results: averaged scores obtained using a VGG16, an Inflated ResNet34, as well as two-channel models trained on four aforementioned tasks. We report results using 1 training sample per class, 5 training samples per class or the full training set.

3. *Effect of the action granularity*: Fig. 5 supports the contention that training on fine-grained tasks yields better features. The best model on this benchmark is that trained jointly on full captions and action categories. The only exception is the model trained to do pure captioning.

# 7 CONCLUSIONS

Pre-training neural networks on large labeled datasets has become a driving force in deep learning applications. Some might argue that it may be considered a serious competitor to unsupervised learning as a means to generate universal features for the visual world. Ever since ImageNet became popular as a generic feature extractor, a hypothesis been that it is dataset size, the amount of detail and the variety of labels that drive a network's capability to learn useful features. To the degree that this hypothesis is true, generating visual features capable of transfer learning should involve source tasks that (i) are fine-grained and complex, and (ii) involve video not still images, because video is a much more fertile domain for defining complex tasks that represent aspects of the physical world.

This paper provides further evidence for that hypothesis, showing that task granularity has a strong influence on the quality of the learned features. We also show that captioning, which to the best of our knowledge has been used only as a *target* task in transfer learning, can be a powerful source task. Our work suggests that one gets substantial leverage by utilizing ever more fine-grained recognition tasks, represented in the form of captions, possibly in combination with question-answering. Unlike the current trend of engineering neural networks to perform bounding box generation, semantic segmentation, or tracking, the appeal of fine-grained textual labels is that they provide a simple homogeneous interface. More importantly, they may provide "just enough" localization and tracking capability to solve a wide variety of important tasks, without allocating valuable compute resources to satisfy intermediate goals at an accuracy that these tasks may not actually require.

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

## APPENDIX

In this supplementary document, we provide:

- Qualitative examples for our classification and captioning.
- Visualization of our classification and captioning models using Grad-CAM.
- The full list of action categories of 20bn-kitchenware.

### QUALITATIVE EXAMPLES OF CLASSIFICATION

Here we provide video examples and their ground truth action categories, along with model predictions for each. We use our M(256-256) which is trained with $\lambda = 0.1$. Interestingly, notice that even when the predicted actions are incorrect, e.g. row 4 in Figure 6, they are, nevertheless, usually quite sensible.

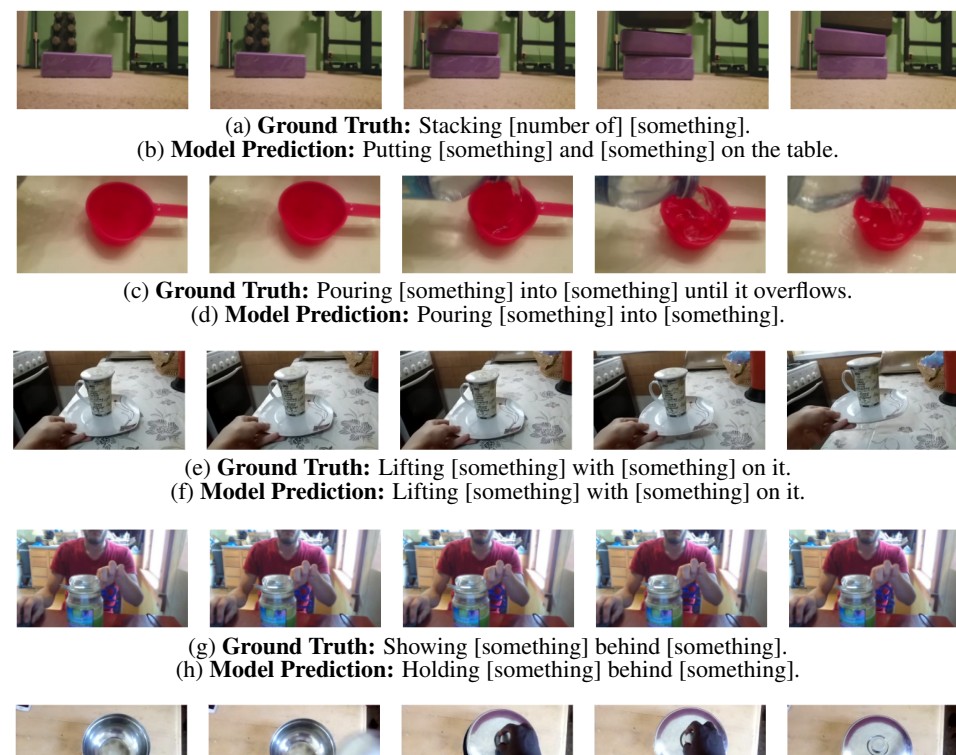

(a) **Ground Truth:** Stacking [number of] [something].
(b) **Model Prediction:** Putting [something] and [something] on the table.

(c) **Ground Truth:** Pouring [something] into [something] until it overflows.
(d) **Model Prediction:** Pouring [something] into [something].

(e) **Ground Truth:** Lifting [something] with [something] on it.
(f) **Model Prediction:** Lifting [something] with [something] on it.

(g) **Ground Truth:** Showing [something] behind [something].
(h) **Model Prediction:** Holding [something] behind [something].

(i) **Ground Truth:** Putting [something] onto [something].
(j) **Model Prediction:** Covering [something] with [something].

Figure 6: Ground truth and model prediction for classification examples.

QUALITATIVE EXAMPLES OF CAPTIONING

Below are examples of videos. accompanied by their their ground truth caption and the caption generated by the model. We use model M(256-256) in this section as well, which is also trained jointly for classification and captioning (with $\lambda = 0.1$).

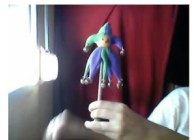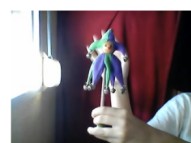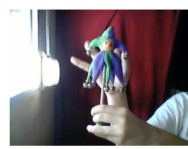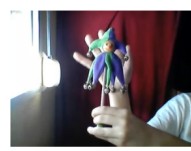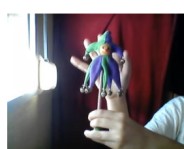

(a) **Ground Truth:** Touching (without moving) the head of a toy.
(b) **Model output:** Poking a stuffed animal so lightly that it doesnt or almost doesnt move.

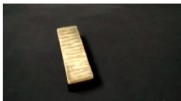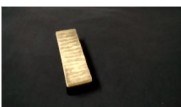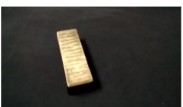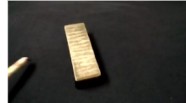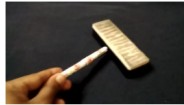

(c) **Ground Truth:** Pushing duster with white coloured pen.
(d) **Model output:** Pushing phone with pen.

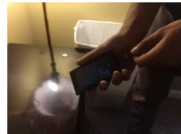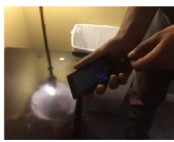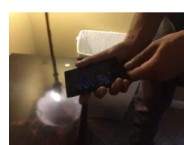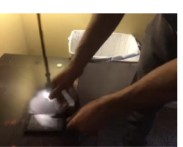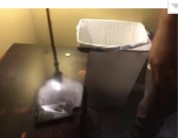

(e) **Ground Truth:** Plugging a charger into a phone.
(f) **Model output:** Plugging charger into phone.

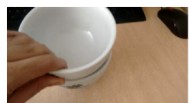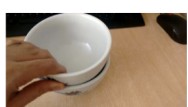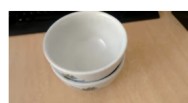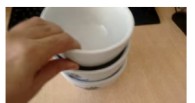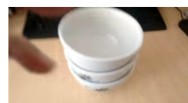

(g) **Ground Truth:** Piling bowl up.
(h) **Model output:** Stacking bowls.

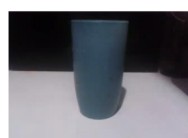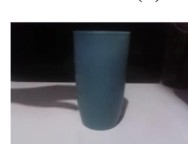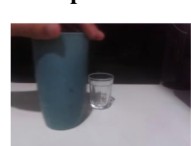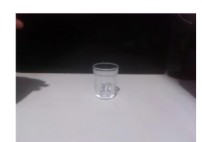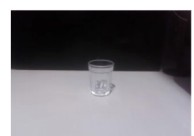

(i) **Ground Truth:** Removing cup, revealing little cup behind.
(j) **Model output:** Removing mug, revealing cup behind.

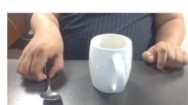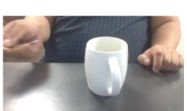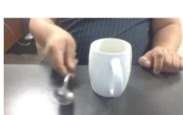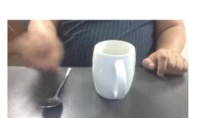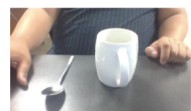

(k) **Ground Truth:** Hitting cup with spoon.
(l) **Model output:** Hitting mug with spoon.

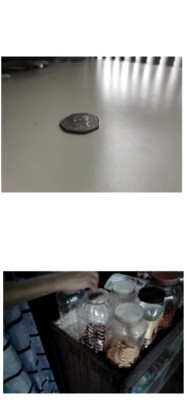 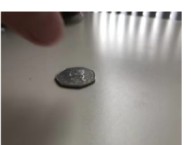 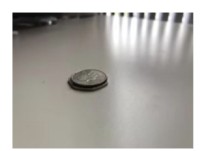 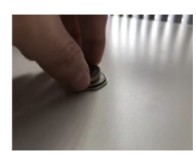 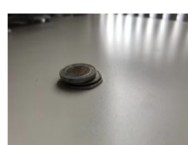

(a) **Ground Truth:** Stacking 4 coins.
(b) **Model output:** Piling coins up.

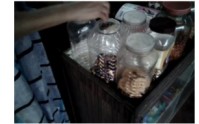 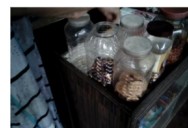 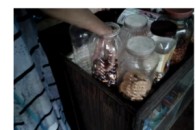 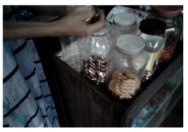 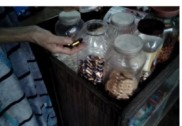

(c) **Ground Truth:** Taking toffee eclairs from jar.
(d) **Model output:** Taking battery out of container.

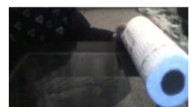 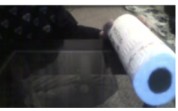 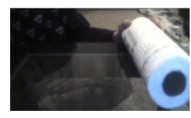 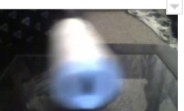 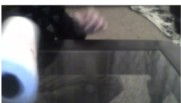

(e) **Ground Truth:** Rolling paper towels on a flat surface.
(f) **Model output:** Letting bottle roll along a flat surface.

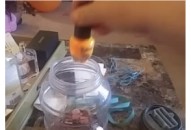 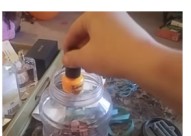 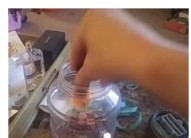 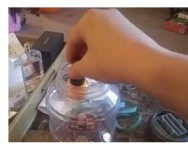 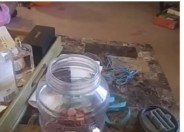

(g) **Ground Truth:** Pretending to put nail polish into jar.
(h) **Model output:** Pretending to put bottle into container.

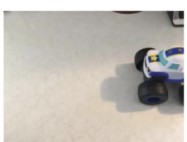 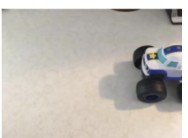 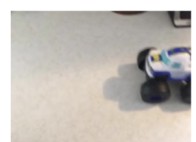 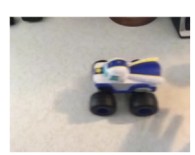 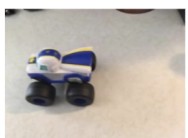

(i) **Ground Truth:** Letting toy truck roll along a flat surface.
(j) **Model output:** Pushing car from right to left.

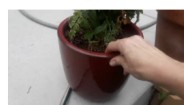 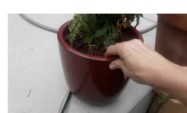 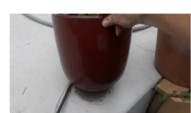 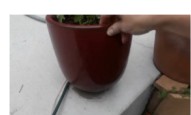 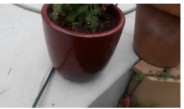

(k) **Ground Truth:** Lifting up one end of flower pot, then letting it drop down.
(l) **Model output:** Lifting up one end of bucket, then letting it drop down.

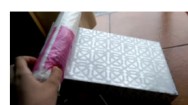 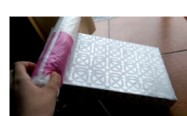 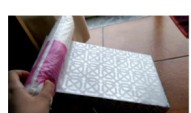 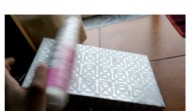 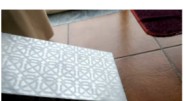

(m) **Ground Truth:** Letting roll roll down a slanted surface.
(n) **Model output:** Letting spray can roll down a slanted surface.

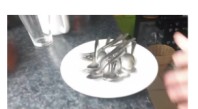 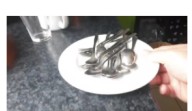 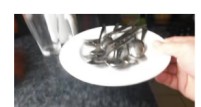 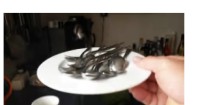 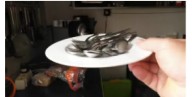

(o) **Ground Truth:** Lifting plate with cutlery on it.
(p) **Model output:** Lifting plate with spoon on it.

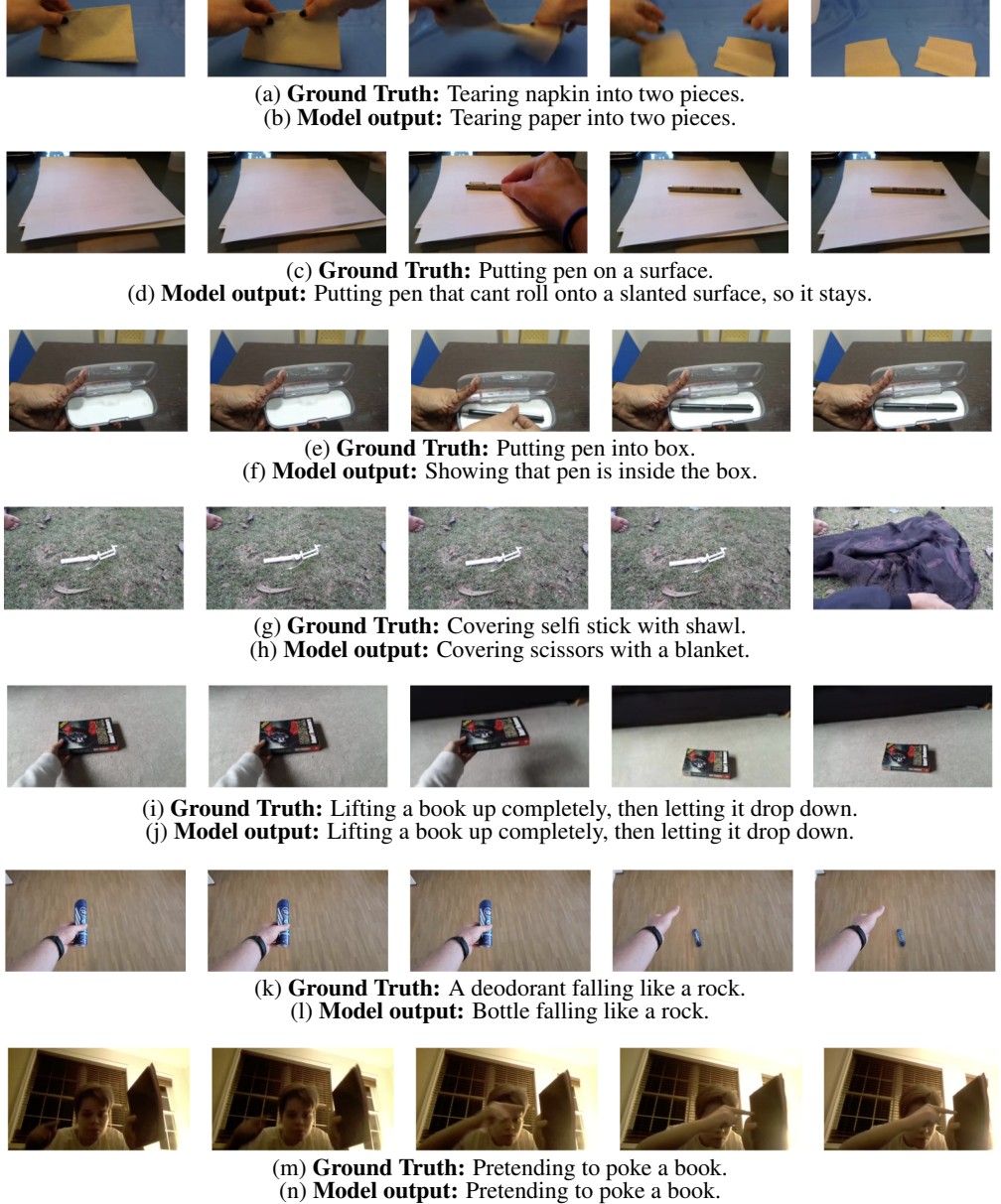

(a) **Ground Truth:** Tearing napkin into two pieces.
(b) **Model output:** Tearing paper into two pieces.

(c) **Ground Truth:** Putting pen on a surface.
(d) **Model output:** Putting pen that cant roll onto a slanted surface, so it stays.

(e) **Ground Truth:** Putting pen into box.
(f) **Model output:** Showing that pen is inside the box.

(g) **Ground Truth:** Covering selfi stick with shawl.
(h) **Model output:** Covering scissors with a blanket.

(i) **Ground Truth:** Lifting a book up completely, then letting it drop down.
(j) **Model output:** Lifting a book up completely, then letting it drop down.

(k) **Ground Truth:** A deodorant falling like a rock.
(l) **Model output:** Bottle falling like a rock.

(m) **Ground Truth:** Pretending to poke a book.
(n) **Model output:** Pretending to poke a book.

Figure 9: Ground truth captions and model outputs for video examples.

## VISUALIZATION OF CLASSIFICATION MODEL WITH GRAD-CAM

To visualize regularities learned from data, we extracted temporally-sensitive saliency maps using Grad-CAM Selvaraju et al. (2016), for both classification and captioning task. To this end we extended the Grad-CAM implementation for video processing. Figure 10 shows saliency maps of examples from Something-Something obtained with model M(256-0) trained on fine-grained action categories, with $\lambda = 1$ (i.e., the pure classification task).

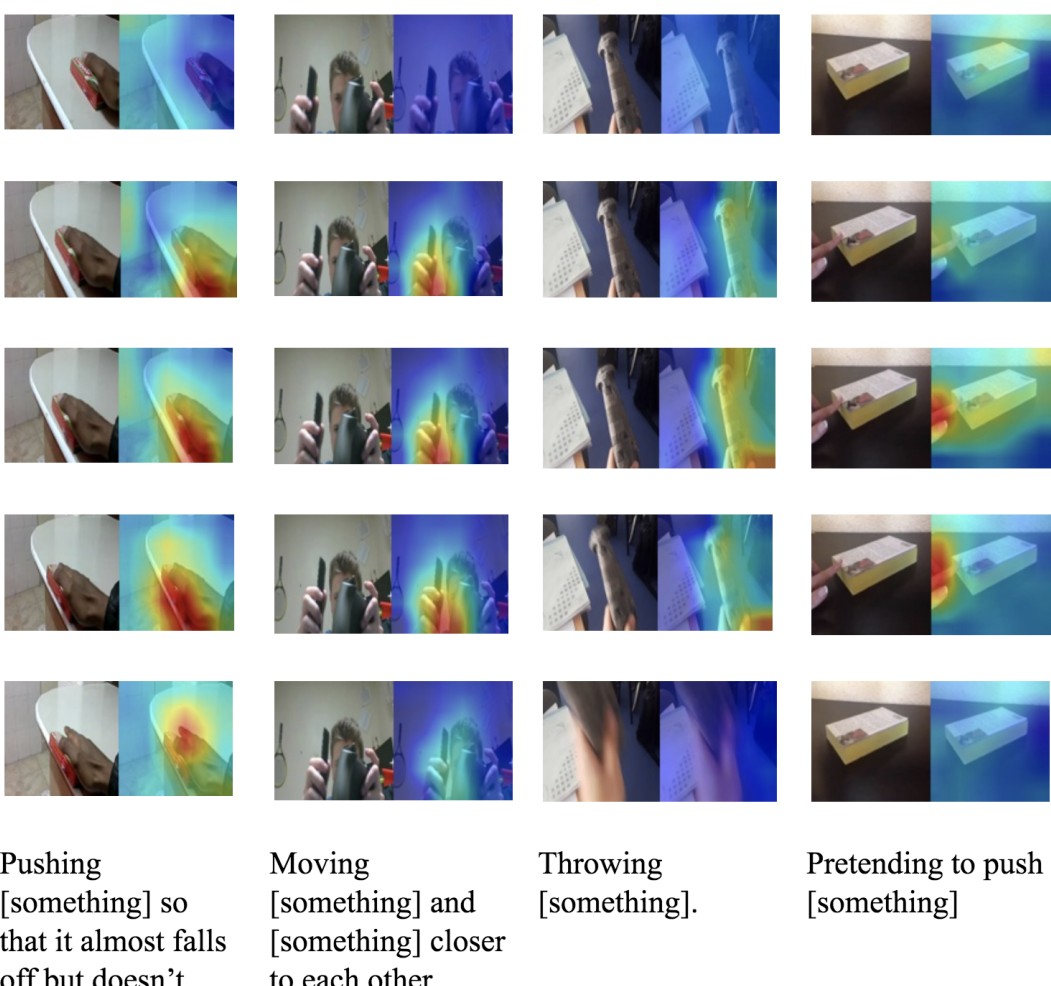

Pushing [something] so that it almost falls off but doesn't.

Moving [something] and [something] closer to each other.

Throwing [something].

Pretending to push [something]

Figure 10: Grad-CAM for M(256-0) on video examples predicted correctly during fine-grained action classification. We can see that the model focuses on different parts of different frames in the video in order to make a prediction.

## VISUALIZATION OF CAPTIONING MODEL USING GRAD-CAM

To get saliency maps during the captioning process, we calculate the Grad-CAM once for each token, for which different regions of the video are highlighted. Figures 11,-13 shows saliency maps for the captioning model, jointly trained with $\lambda = 0.1$. Notice how the attentional focus of the model changes qualitatively as we perform Grad-CAM for different tokens in the target caption.

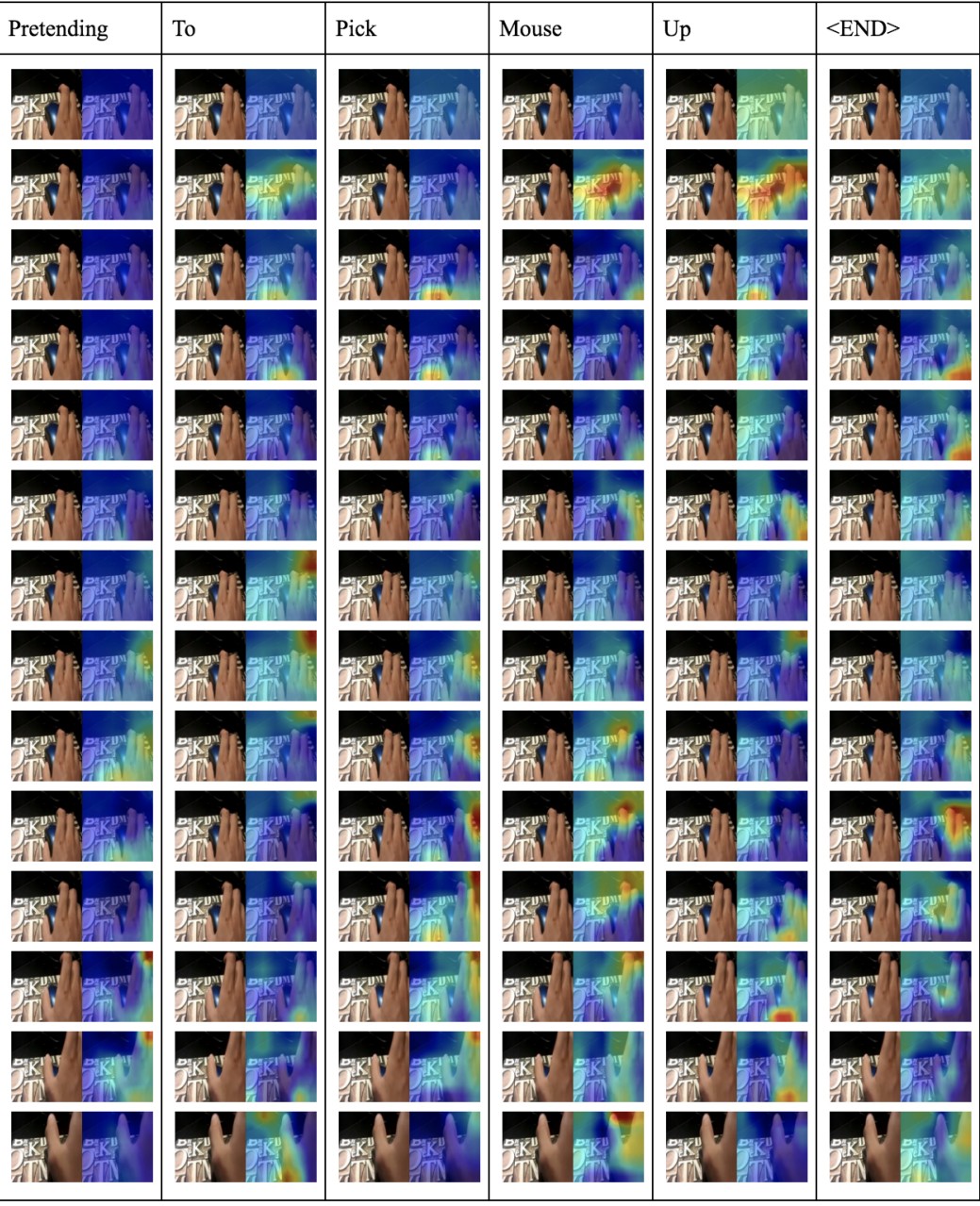

Figure 11: Grad-CAM on video example with ground truth caption **Pretending to pick mouse up.** The model focuses on hand motion in the beginning and end of the video for the token "Up".

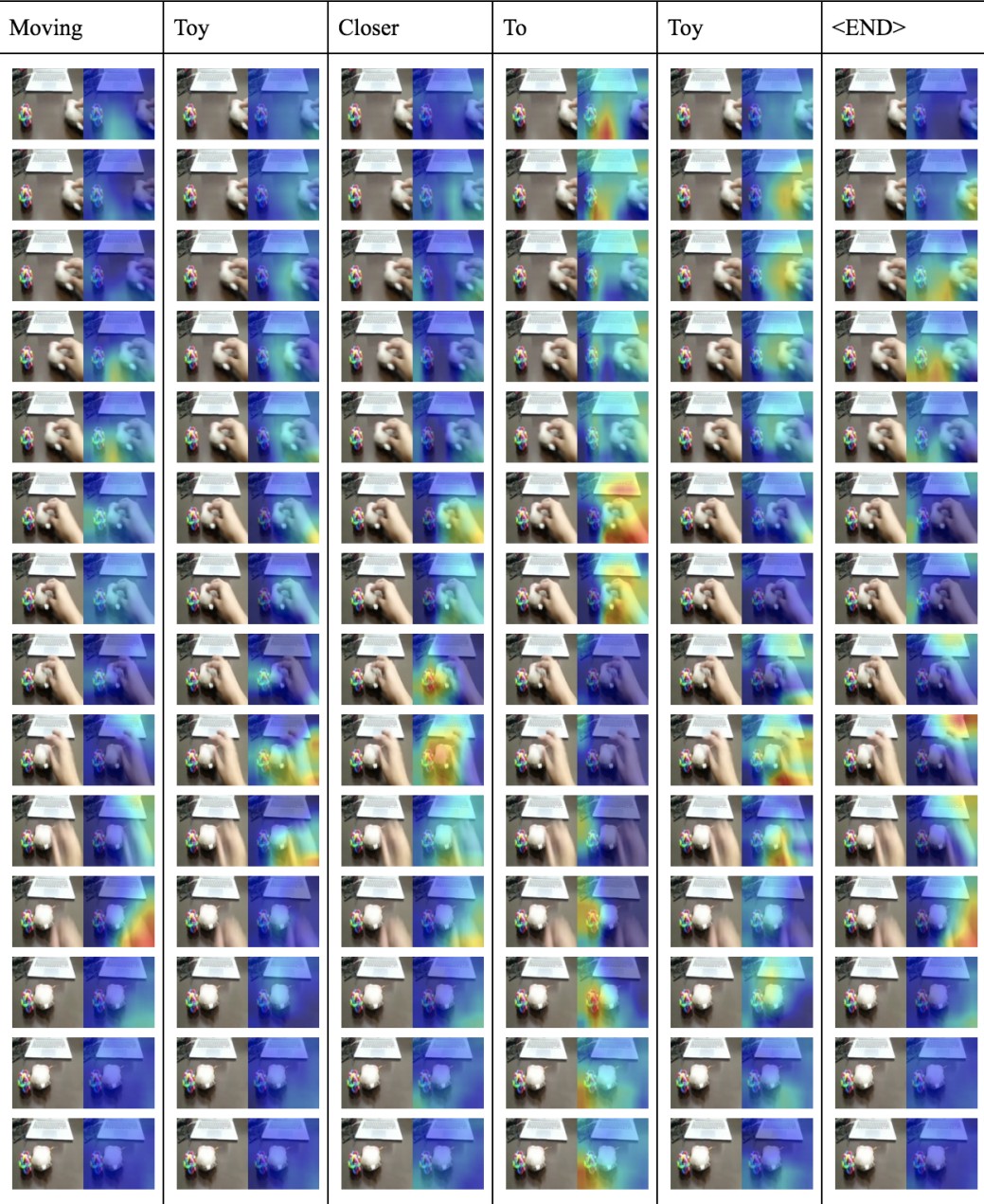

Figure 12: Grad-CAM on video example with ground truth caption **Moving toy closer to toy.** We can see that the model focuses on the gap between toys when using "Moving" token. It also looks at both toy objects when using the token "Closer".

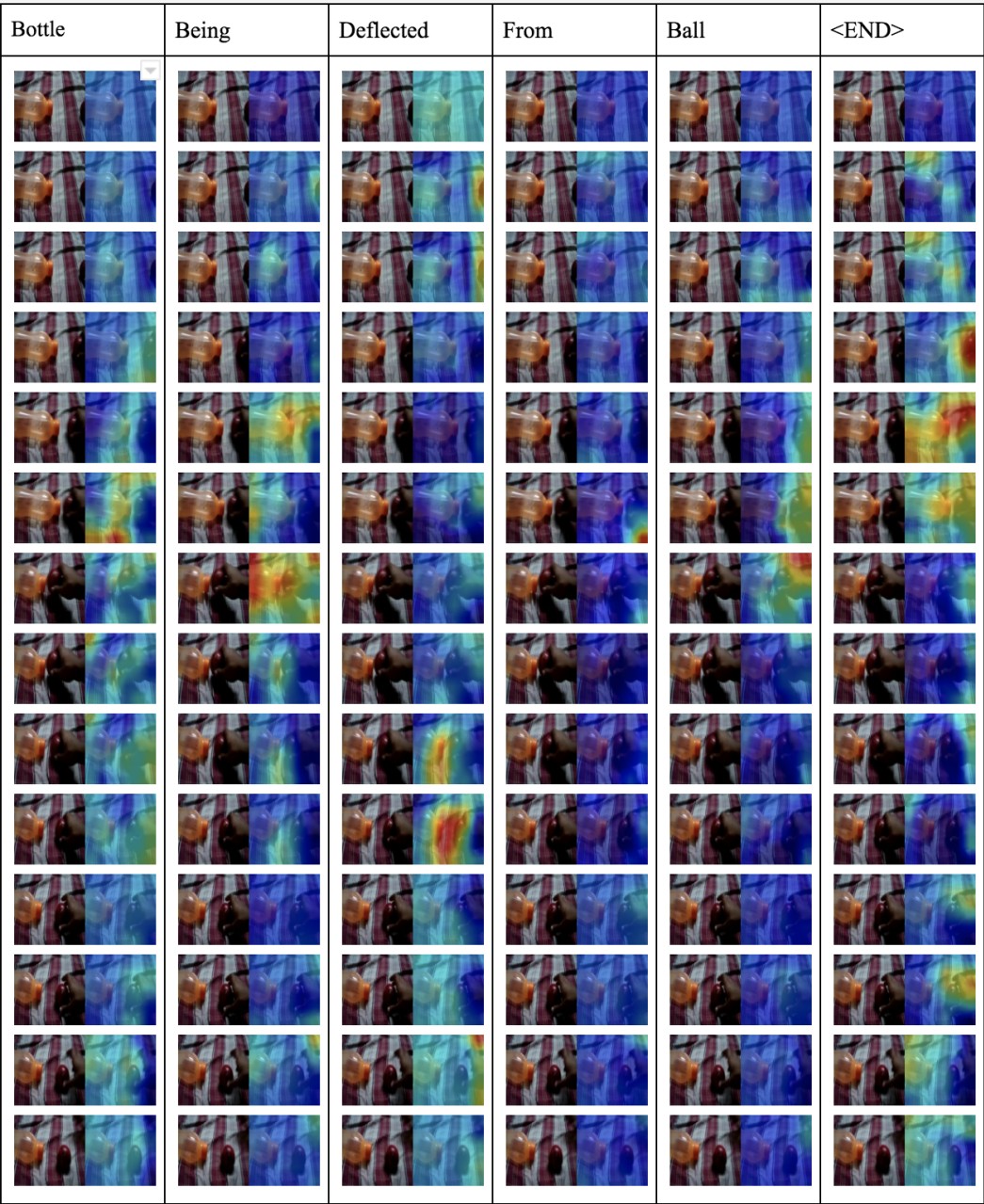

Figure 13: Grad-CAM on video example with ground truth caption **Bottle being deflected from ball** during captioning process. The model focuses on the collision between bottle and ball, when using token "Deflected".

20BN-KITCHENWARE ACTION CATEGORIES

Table 9 lists the full list of 20bn-kitchenware action categories.

| Action categories |
|---|
| Using a fork to pick something up |
| Pretending to use a fork to pick something up |
| Trying but failing to pick something up with a fork |
| Using a spoon to pick something up |
| Pretending to use a spoon to pick something up |
| Trying but failing to pick something up with a spoon |
| Using a knife to cut something |
| Pretending to use a knife to cut something |
| Trying but failing to cut something with a knife |
| Using tongs to pick something up |
| Pretending to use tongs to pick something up |
| Trying but failing to pick something up with tongs |
| Doing other things |

Table 9: The 13 action categories represented in 20bn-kitchenware.

