# OpenReview forum: "ON THE EFFECTIVENESS OF TASK GRANULARITY FOR TRANSFER LEARNING"
_ICLR.cc/2019/Conference_

### Official Review · AnonReviewer3 · 2018-11-02
**Too many results without proper analysis.**

**Rating:** 5
**Confidence:** 4

**Review:**

Paper Summary - This paper presents an approach for fine-grained action recognition and video captioning. The authors train a model using both classification and captioning tasks and show that this improves performance on transfer learning tasks. The method is evaluated on the Something-Something v2 dataset as well as a new dataset (proposed in this paper). The authors also evaluate the benefit of using fine-grained action categories vs. coarse-grained action categories on transfer learning.

Paper Strengths
-  Comparing fine-grained vs. coarse-grained action categories for transfer learning is well motivated. Evaluating just this aspect in the context of video classification is helpful (Section 5.1). Establishing the baseline using linear classifiers for feature transfer makes the feature transfer result more robust. The authors have also done a good job of evaluating their method in the coarse-grained and fine-grained settings (Table 1, 2).
- The architectural and experimental design in this paper is well illustrated.
- The 20bn kitchen dataset has interesting categories about intention - pretending to use, using, and using & failing.
- The ablation in Table 1 is helpful in understanding the contribution of 3D vs. 2D convolutions.

Paper Weaknesses
- I believe this paper tries to do too much and as a result fails to show results convincingly. There are too many results and not much focus on analyzing them. In my opinion, the experimental setup in the paper is weak to fully support the authors' claims.
- I now analyze the main contributions of this paper as outlined by the authors in Section 1.
    - Label granularity and feature quality: To me this is the most interesting part of this paper and most related to its title. However, this is also the most under-analyzed aspect. The only result that the authors show is in Sec 5.1 and Fig 5. Apart from using the provided fine-grained vs. coarse-grained labels for evaluation, the authors do not perform many experiments in this domain and neither do they analyze these results. For example, the gain using fine-grained labels is not significant in Figure 5 (2Channel - CG vs. 2Channel - FG). The authors do not explain this aspect. Another missing baseline from Figure 5 is "2Channel - Captions & CG actions". This baseline is needed to understand the contribution of FG vs CG actions when also using captioning as additional supervision.
    - Baselines for captioning: The authors do not provide any details for this task. If the intent is to establish baselines there needs to be more effort on analyzing design decisions - e.g. decoding, layers in LSTM. Captioning metrics such as CIDER and SPICE are missing.
    - Captions as a source for transfer learning: This is poorly analyzed in this paper. 1) Can the captions be converted to "tags" and then used for supervision? What is the benefit of producing the full sequential text description over this simple approach? 2) Captions for transfer learning are only analyzed in Figure 5 without much explanation. It is hard to claim that captioning is the reason for performance gains without really analyzing it completely.
    - 20bn-kitchenware dataset - This dataset is explained in just one paragraph in Section 6. What is the motivation behind collecting this dataset as opposed to showing transfer learning on some other dataset?
- Missing references
        - There has been work in understanding the effect of fine-grained categories in ImageNet transfer learning - What makes ImageNet good for transfer learning? Huh et al. What is the insight provided over this work?
- Minor comments
    - Section 1: Figure 4 is referenced in points 1 & 3. I think you mean Figure 5.

---

### Official Review · AnonReviewer2 · 2018-11-02
**Interesting observation and dataset, but more analysis would be necessary**

**Rating:** 5
**Confidence:** 4

**Review:**

Summary
This paper studied video classification and video caption generation problem.
Especially, the paper tried few baseline architectures datasets used from pretraining features on recently proposed Something-something-V2 dataset and another newly proposed dataset.
This paper argues that fine-grained annotation helps learning good features and enhance transfer learning results

Strength
There are some interesting observations in terms of transfer learning.
Especially, comparison of fine-grained and coarse-grained dataset for transfer learning (Table 2), and effect of using caption and newly collected dataset for transfer learning (Figure 5) is interesting and the result is worth to be shared to the community.
In addition, a new dataset that are carefully collected for transfer learning might be useful to make progress on video classification and captioning.

Weakness
To many tables with different neural network parameter settings look distracting and does not provide much information. Instead, focusing more on effect of dataset for transfer learning and providing more analysis on this aspect would make the main argument of this paper stronger.
For example, effect of transfer could be studied on different dataset. If transfer learning with proposed dataset containing find-grained annotation / captions is useful, it might help boosting performance on other video recognition dataset as well.
Providing analysis on understanding the effect of fine-grained / captioning dataset for feature learning might help understanding as well.

Overall rating
This paper suggest interesting observations and useful dataset, but provides relatively less analysis on these observations. I believe providing more analysis on the dataset and effect of transfer would make the main argument of this paper stronger.

---

### Official Review · AnonReviewer1 · 2018-11-02
**Nice paper but somewhat limited novelty out of the specific video classification & captioning domain considered**

**Rating:** 5
**Confidence:** 4

**Review:**

This paper describes a multi-task video classification and captioning model applied to a fine-grained object relationship video dataset, for a range of different classification and captioning tasks at different levels of granularity. This paper also creates a new video action dataset around kitchen objects and actions.  Finally, the paper includes an empirical study on both the multi-task performance and transfer learning performance between the two datasets considered.

Pros:
- This paper is clearly written and includes a thorough and well-laid out empirical component
- The contribution to the video action classification and captioning space seems like a worthwhile one

Cons:
- The novelty of this paper mainly seems to be with respect to video classification and captioning; other methodological aspects and empirical themes are interesting but fairly standard more generally.  The lack of experiments outside of one video action classification & captioning dataset (and one additional one for a transfer learning study) limit the empirical generality of the findings.

Overall take: This paper's contributions seem of interest to the video classification and captioning community, but less so to a broader or more methodologically-focused one such as ICLR.

Notes:
- The comments on insufficiency of existing video classification tasks in Sec. 3 are interesting, but seem pretty restricted to that specific domain
- The model used is a fairly standard CNN + LSTM video encoder, plus a basic MTL network approach with hard parameter sharing between tasks, as is commonly used today. Similarly, the transfer learning approach---pre-training on one task, then freezing layers and fine-tuning---is a standard approach.
- The empirical findings are interesting---for example, that training on fine-grained tasks improves coarse-grained accuracy, that MTL training is helpful, etc---but (a) seem in general like known themes, and (b) have limited generality either way beyond the specific types of tasks considered in the dataset examined.
- In general, much of the paper is focused on details specific to this application domain, rather than to general methods or themes potentially interesting to the broader ICLR community

---

### Meta-Review · Area_Chair1 · 2018-12-14
**Relatively weak novelty and empirical results.**

**Confidence:** 4
**Recommendation:** Reject

**Metareview:**

This paper presents the empirical relation between the task granularity and transfer learning, when applied between video classification and video captioning. The key take away message is that more fine-grained tasks support better transfer in the case of classification---captioning transfer on 20BN-something-something dataset.

Pros:
The paper presents a new empirical study on transfer learning between video classification and video captioning performed on the recent 20BN-something-something dataset (220,000 videos concentrating on 174 action categories). The paper presents a lot of experimental results, albeit focused primarily on the 20BN dataset.

Cons:
The investigation presented by this paper on the effect of the task granularity is rather application-specific and empirical. As a result, it is unclear what generalizable knowledge or insights we gain for a broad range of other applications. The methodology used in the paper is relatively standard and not novel. Also, according to the 20BN-something-something leaderboard (https://20bn.com/datasets/something-something), the performance reported in the paper does not seem competitive compared to current state-of-the-art. There were some clarification questions raised by the reviewers but the authors did not respond.

Verdict:
Reject. The study presented by the paper is a bit too application-specific with relatively narrow impact for ICLR. Relatively weak novelty and empirical results.